# On-Line Preconcentration and Simultaneous Determination of Cu and Mn in Water Samples Using a Minicolumn Packed with Sisal Fiber by MIP OES

**DOI:** 10.3390/molecules26061662

**Published:** 2021-03-16

**Authors:** Javier Silva, Mariela Pistón

**Affiliations:** 1Graduate Program in Chemistry, Facultad de Química, Universidad de la República, Avda. Gral. Flores 2124, Montevideo 11200, Uruguay; jsilva@fq.edu.uy; 2Grupo de Análisis de Elementos Traza y Desarrollo de Estrategias Simples para Preparación de Muestras (GATPREM), Química Analítica (DEC), Facultad de Química, Universidad de la República, Montevideo 11200, Uruguay

**Keywords:** biosorbents, sisal fiber, MIP OES, water, preconcentration, trace element, sample preparation

## Abstract

An on-line preconcentration system for the simultaneous determination of Copper (Cu) and manganese (Mn) in water samples was developed and coupled to a microwave-induced plasma optical emission spectrometer (MIP OES). The flow injection system was designed with a minicolumn packed with sisal fiber (*Agave sisalana*). A multivariate experimental design was performed to evaluate the influence of pH, preconcentration time, and eluent concentration. Optimal conditions for sample preparation were pH 5.5, preconcentration time was 90 s, and HCl 0.5 mol L^−1^ was the eluent. The main figures of merit were detection limits 3.7 and 9.0 µg L^−1^ for Cu and Mn, respectively. Precision was expressed as a relative standard deviation better than 10%. Accuracy was evaluated via spiked recovery assays with recoveries between 75–125%. The enrichment factor was 30 for both analytes. These results were adequate for water samples analysis for monitoring purposes. The preconcentration system was coupled and synchronized with the MIP OES nebulizer to allow simultaneous determination of Cu and Mn as a novel sample introduction strategy. The sampling rate was 20 samples/h. Sisal fiber resulted an economical biosorbent for trace element preconcentration without extra derivatization steps and with an awfully time of use without replacement complying with the principles of green analytical methods.

## 1. Introduction

The determination of trace elements in water samples is of interest both to help us understand more about the levels of essential elements, as well as those that are potentially toxic. Copper (Cu) and manganese (Mn) are expected to be found at low levels in natural waters (e.g., well water, rain, river, among other resources), but human activity is also responsible for finding these heavy metals in waters [1,2]. Both can also be present in drinking water due to the corrosion of pipes, the erosion of natural deposits and due to the use of products for purification [3,4]. The United States Environmental Protection Agency (US EPA) provides guidelines for communities on maximum allowable concentration levels (MCL) to avoid health and organoleptic problems due to the presence of these elements (e.g., unpleasant appearance, odor, or taste). The MCLs stablished for Cu and Mn in drinking water are 1 and 0.05 mg L^−1^, respectively [5]. Atomic spectrometry techniques have been widely used for trace element determinations in water samples. Within these techniques, atomic emission is preferred since it allows for a multielement analysis. The use of inductively coupled plasma atomic emission spectroscopy (ICP OES) and inductively coupled plasma mass spectrometry (ICP-MS) can achieve the ionization of elements with a high efficiency, achieving low detection limits (ng L^−1^), so they are the first choice when working at trace and ultra-trace levels [6]. As a disadvantage, they have a high investment cost, maintenance, and a high consumption of argon during operation. As an alternative to these expensive techniques, in recent decades, microwave-induced plasma atomic emission spectrometry (MIP OES) has emerged using a nitrogen plasma. MIP OES appears as a more economical and ecological technology since the plasma is maintained with nitrogen generated from an air compressor (“runs on air”); thus, the operating costs are considerably lower and the technique is considered environmentally friendly. On the other hand, the nitrogen plasma does not reach such high temperatures than the argon one, therefore detection limits are higher (µ L^−1^). This disadvantage is detrimental to its wider application in trace or ultra-trace element analysis [7].

An alternative to overcome this drawback is the introduction of the sample after an online preconcentration step. This can be accomplished by coupling an automated flow injection system (FIA) that allows to increase the amount of sample introduced into the MIP OES plasma and therefore increase the sensitivity. In addition, this strategy can avoid, in many cases, possible interferences produced by the matrix. FIA has several advantages when coupled to atomic spectrometry; however, to reach microscale working conditions with reliable and reproducible results, rigorous optimization is required [8,9,10,11,12,13,14].

Biosorbents are economic and ecological materials that have been successfully used for heavy metal remediation as adsorbents in wastewater [15]. These materials can accumulate metals through metabolic pathways or physicochemical processes. Several reports indicate that some biosorbents are efficient when they have many binding sites such as carboxylic groups, carbonyls, and amines, among others [15,16]. Lignocellulose present in the cell wall of plant material has carboxylic and hydroxyl groups; therefore, materials such as sisal fiber (*Agave sisalana*), algae, or chitinous materials have been tested as environmental pollution remediation agents [15,17]. Based on what has been reported about applications of these materials in bioremediation, it was proposed to test sisal fiber for preconcentration packing minicolumns. Few applications were published and all of them functionalize the sisal fiber with different reagents, for example using alizarin fluoride blue for copper or tiazolylazo-Resorsinol (TAR) for cadmium preconcetration, respectively [16,18]. Other studies used sisal fiber for off-line solid phase extraction of Cu, Mn, Ni, and Zn from diesel oil samples [19].

Compared with commercial solid-phase sorbents, biosorbents are renewable materials with the advantages of being low cost and easily available, characteristics that contribute to the principles of green chemistry [13,20,21].

The application of these greener techniques can be useful for water pollution monitoring and surveillance of the quality of the potable water supplied by distributing stations. This can contribute to the management of adequate water treatment technologies [22].

The goal of this work is the use of sisal fiber as biosorbent for preconcentration of Cu and Mn without need of functioning the material with other reagents using an online flow injection system with a minicolumn. The developed system was coupled to MIP OES after experimental conditions were optimized and validated for the determination of these elements in water samples. The preconcentration and sample introduction steps were automated allowing simultaneous measurement of both elements with good precision using hydrochloric acid 0.5 mol L^−1^ as eluent. Several water samples (well, rain, tap water) were analyzed, and the results were compared with those obtained using electrothermal atomic absorption spectrometry (ETAAS). The developed method result was simple and accurate, being a promising alternative for heavy metal determination using greener analytical techniques.

## 2. Results and Discussion

A flow injection system (FI) was coupled to a microwave induced plasma spectrometer (MIP OES) with an on-line preconcentration system for Cu and Mn determinations. A minicolumn packed with a biosorbent (sisal fiber) was placed in a 6-port valve preconcentation of both elements in water samples to improve the detection limits using an automated process for this purpose. This development involved an optimization of the coupling conditions and the sample preparation step, and, finally, the validation and application to real samples.

### 2.1. Flow Injection System Coupled to MIP OES

Obtaining analytical data with good precision is essential when developing a new analytical method. Particularly when additional steps or modifications are made to conventional mode of operation of the instrument, this parameter must be carefully studied.

Coupling an on-line preconcentration flow system to an atomic emission technique such us MIP OES was a challenge because the elution step required to be synchronized with the detection system. Sample introduction, after elution from the minicolumn, into the nebulizer was a critical stage.

The MIP OES spectrometer (Agilent 4210, the latest model available) operates with a sequential monochromator, i.e., it modifies the reading conditions for each analyte moving the grating sequentially to each selected wavelength. Multielement analysis can be made with this technique in the conventional mode of operation since the grating movement occurs in a few seconds. However, this is a problem when a flow system is coupled because a real-time mode of measurement is required (time scan mode), and the time that the instrument needs to move the grating from a wavelength to another, for multielement analysis, caused severe imprecision. This data acquisition mode entails loss of time which negatively influences the accuracy when performing data acquisition in real-time.

Figure 1 shows the imprecise behavior of the analytical signals (peak height) when a standard solution of Cu and Mn (2 mg L^−1^) was registered in quintuplicate.

Therefore, making the instrumental conditions for both analytes compatible with a simultaneous determination, reducing the loss of time, and thus improving the precision and increasing the sampling frequency was a challenge. To achieve this goal, the beginning of data acquisition was synchronized with the start of the preconcentration step using a program written in Phyton language. Furthermore, to minimize the loss of time between measurements, a compromise was made with the operating conditions of the MIP OES, as shown in Table 1.

### 2.2. Optimization

A multivariate experimental design (three-level central composite) was performed using a standard solution of Cu and Mn (5 mg L^−1^) to optimize critical variables. The effects of pH, preconcentration time, and eluent concentration were evaluated. Each experiment was carried out in triplicate. The experimental conditions and the results are presented in Table 2.

According to the results, the best conditions in terms of better analytical signal responses were achieved for preconcentration times: 90 s, pH: 5.5, and HCl concentration of 0.5 mol L^−1^. Therefore, these conditions were selected for validation (experiment 11).

For each determination, sample consumption was less than 10 mL and 3 mL of 0.5 mol L^−1^ HCl for the elution step.

### 2.3. Validation

The analytical performance of the complete process, on-line preconcentration, and subsequent determination of Cu and Mn in water samples was evaluated in accordance with the Eurachem Guide recommendations. The detection and quantification limits were obtained according to the 3 *s* and 10 *s* criteria [23]. Trueness was evaluated using a spike-recovery assay; this was because available certified reference materials were preserved in an acid medium. This acidic medium implies a strong pH adjustment that involves a large amount of sodium hydroxide that had negative effects in the preconcentration step when it was tested. Adding more sodium hydroxide was also incompatible with the amount of total dissolved solids recommended for the MIP OES technique (up to 2%). Therefore, the spike-recovery (%R) assay was performed at two concentration levels, 0.1 and 0.2 mg L^−1^ for Cu an Mn, respectively.

Precision was expressed as relative standard deviation (RSD%) and was evaluated using a pool of three water samples spiked with 0.05 and 0.1 mg L^−1^ of Cu and Mn respectively (n = 6) and with samples that quantified Cu and Mn. Linearity was good for both elements with determination coefficients (R^2^) greater than 0.998. Table 3 summarizes the obtained figures of merit of the method validation.

These results are suitable for Cu and Mn monitoring purposes considering the MCL values for both metals in waters. Linear ranges were 10 times wider than those obtained using electrothermal atomic absorption spectrometry (ETAAS) when applied in this work for comparison of the samples results (50 and 15 µg L^−1^ for Cu and Mn respectively) and in previous works [24].

When precision was optimized before the synchronization of RSD (%) was close to 20% for both elements, the improvement was notable when this limitation was solved, achieving values of RSD better than 5% in spiked samples at low concentration levels (Table 3). Figure 2 shows the recording of the signal, simultaneously, for Cu and Mn, using the time scan mode for three standards and three samples each in triplicate. Thus, the improvement of the precision can be observed.

The enrichment factor (EF) is an important figure of merit for preconcentration systems. For both analytes this factor was 30 with a sampling frequency of about 20 samples h^−1^ for the simultaneous determination of Cu and Mn. More than 350 determinations have been made in this work using the same package in the minicolumn without regeneration; no signal decay was observed. Another advantage of the FI system was the low consumption of reagents with the consequent low generation of waste.

LOD and LOQ were greater than those that achieved by ETAAS or ICP techniques. However, considering the MIP OES capabilities and the fit-for-purpose, these results can be considered adequate.

Commercial resins or sorbents are commonly used for preconcentration purposes, but compared to commercial sorbents, biosorbents have the advantages of being low cost and highly available renewable materials, attributes that make them a promising eco-alternative for preconcentration.

The validated FI-MIP OES method was applied for the determination of Cu and Mn in the water samples.

The obtained results were statistically compared with those obtained by electrothermal atomic absorption spectrometry (ETAAS). Table 4 summarizes the results.

The samples that presented quantifiable values of Cu and Mn were statistically comparable with the results obtained via ETAAS, a technique widely used for the determination of trace elements in water. Therefore, the proposed analytical method can be postuled as an alternative with several advantages in terms of green analytical chemistry.

## 3. Materials and Methods

### 3.1. Reagents and Instruments

Cooper and Mn calibration standards were prepared from a 1000 mg L^−1^ commercial solution (Fluka, Switzerland) by appropriate dilutions using ultrapure water and pH = 5.5 was adjusted with a diluted dissolution of NaOH (0.1 mol L^−1^).

Purified water (resistivity 18.2 MΩ.cm) was obtained by a purification system (Millipore, DirectQ3 UV, Darmstadt, Germany). All other chemical regents were of analytical grade. All glassware was soaked overnight in 10% (*v*/*v*) nitric acid and then rinsed with ultrapure water.

The flow injection (FI) manifold used for the on-line preconcentration consisted of the following components: a peristaltic pump (Rainin Dynamax RP-1, Emeryville, CA, USA) fitted with Tygon tubing and a 6-port injection valve (two-position) with a microelectronic actuator-controlled module (VICI, Valco Cheminert Instrument Co Inc., Houston, TX, USA), this valve was controlled by a personal computer via the RS232 serial port using the program Phyton™. Connections were made from Teflon PFA tubing (0.8 mm internal diameter). A scheme of the FI system is presented in Figure 3.

The system was operated using a program, created in our laboratory, and written in Phyton™ [25], a general purpose programming language. This program allowed us to control the injection valve and, therefore, the preconcentration and elution steps automatically.

Analytical determinations were carried out using a microwave-induced plasma optical emission spectrometer (MIP OES, Agilent 4210, Santa Clara, CA, USA) with a standard torch, an inert One Neb nebulizer, and a glass cyclonic spray chamber. The instrument works with nitrogen generated with an Agilent 4107 Nitrogen generator (Agilent Technologies, Santa Clara, CA, USA), which works with an air compressor model KK70 TA-200 K (Dürr Technik, Bietigheim-Bissingen, Germany). The plasma operation conditions were gas flow fixed at 20 L min^−1^ and the auxiliary gas flow at 1.5 L min^−1^.

Analytical signals were obtained in Time Scan mode at 324.75 nm and 403.08 nm for Cu and Mn, respectively, generated data was exported in csv format, and processed using the Peak Simple™ software (SRI, Torrance, CA, USA). This software provided tools for signal smoothing, baseline correction, and peak-height (analytical signal) measurement.

For validation and comparison of the obtained results, real samples were also analyzed via electrothermal atomic absorption technique (ETAAS). A longitudinal heated, ETAAS spectrometer (Perkin Elmer HGA 900, Shelton, CT, USA) with hollow cathode lamps (Perkin Elmer, Shelton, CT, USA) of Cu and Mn operating at 324.8 nm and 279.51 nm, respectively, was used. Atomization took place on solid pyrolytic graphite L’vov platform inserted in standard pyrocoated graphite tubes. The programs used for these determinations are shown in Table 5, volume of injection was 0.050 mL, and no matrix modifier was required.

### 3.2. Preconcentration Minicolum

A minicolumn was designed and constructed in the laboratory. It was prepared by packing 100 mg of sisal fiber (Agave sisalana) into a glass tube 5 cm long and 2.2 mm internal diameter. A picture of the packed column is shown in Figure 4, the connections were made by 3D impression. To avoid loss of packing during analysis, a frit was placed at each end of the column.

Sisal fiber was obtained in a local market as sisal thread, before packing the minicolumn the fiber was treated as previously reported by dos Santos et al. [21]. The fibers were firstly washed with HNO_3_ (10% *v*/*v*) and after that, washed successive times with ultrapure water. It was then placed in an oven at 60 °C for 12 h before packing the minicolumn [21].

The operation of the FI system for Cu and Mn simultaneous determination consisted of a preconcentration of the sample or standards in the minicolumn for 90 s. After that, by switching the valve (V) for another 90 s (dotted line in Figure 3), the eluent (HCl 0.5 mol L^−1^) releases the analytes and carries them to the MIP OES. The HCl is propelled by a peristaltic pump included in the MIP OES spectrometer that introduce the sample directly into the nebulizer.

The preconcentration step (switch of the valve and timing) was controlled with a computer with a developed program using Phyton language. The software also controlled the number of repetitions of the whole process for each sample or standards. Thus, this can be considered an automated system for sample preparation and an introduction into the plasma.

### 3.3. Samples

Samples were selected to obtain different representative matrices to apply the developed method.

Water samples were obtained from different regions of Uruguay including six well waters, rainwater, and tap water. The initial pH of the collected samples ranged 5 to 8 and were stored between 2–8 °C in plastic bottles without adding extra preserves until analysis.

Prior to the analytical determinations, if samples presented material in suspension it was filtered through a 0.45 µm pore size membrane. This step was carried out only when necessary. In all cases, the pH was adjusted to 5.5 with a diluted sodium hydroxide solution (0.1 mol L^−1^) or diluted hydrochloric acid (0.1 mol L^−1^), as appropriate. The final volume was established gravimetrically after this addition.

### 3.4. Optimization and Validation

A multivariate experimental was performed to evaluate the influence of pH, preconcentration time and eluent concentration, by means of a three-level central composite design. Peak height was the response evaluated to obtain the optimal conditions for the preconcentration system [26].

The analytical methodology was validated in accordance with the recommendations established via the Eurachem Guide, obtaining the following figures of merit: linearity, detection/quantification limits (LOD and LOQ), precision, and trueness [23].

## 4. Conclusions

A novel on-line preconcentration flow system that uses a minicolumn packed with a biosorbent was developed. The analytical method succeeded in synchronizing the analytical signals of Cu and Mn to record real-time measurements via MIP OES with adequate precision. The use of sisal fiber as a biosorbent was very efficient with up to 350 determinations before requiring replacement and no additional reagent was required to functionalize it.

The figures of merit were adequate for Cu and Mn monitoring purposes in water samples according to international regulations. It was applied to several waters with results that were statistically comparable to those obtained using ETAAS. This development is original in analytical chemistry and has several advantages such as being economical and environmentally friendly.

## Figures and Tables

**Figure 1 molecules-26-01662-f001:**
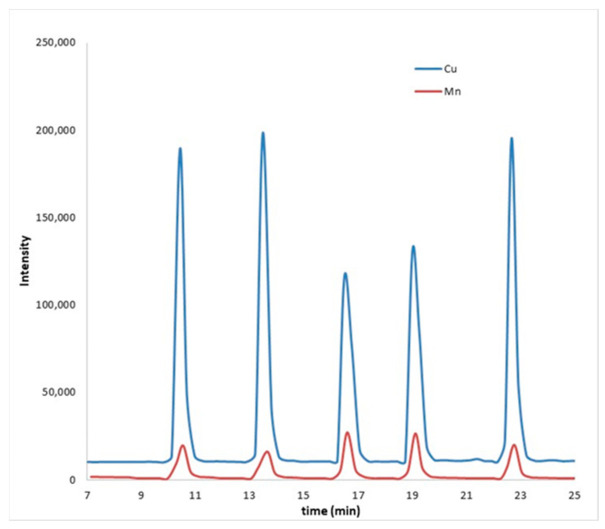
Recording of the signal corresponding to a standard solution (2 mg L^−1^) of Cu (blue) and Mn (red).

**Figure 2 molecules-26-01662-f002:**
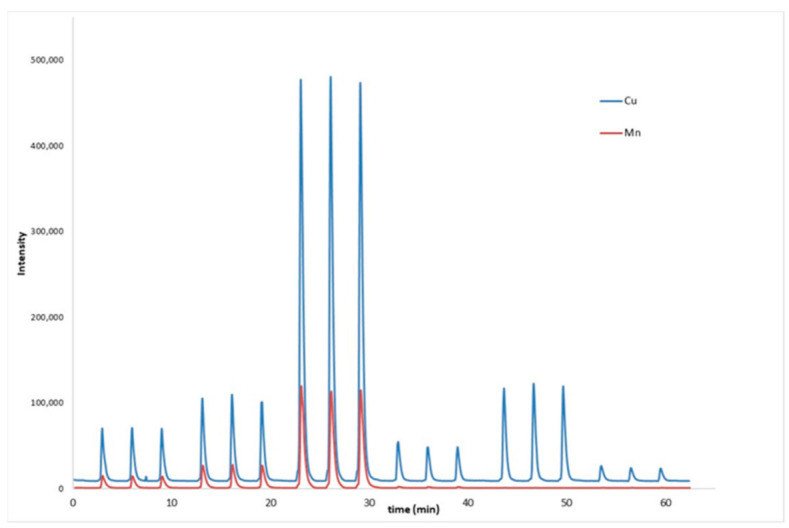
Recording of the signal corresponding to standards (0.05, 0.1, and 0.5 mg L^−1^) and samples with synchronization. Cu (blue), Mn (red).

**Figure 3 molecules-26-01662-f003:**
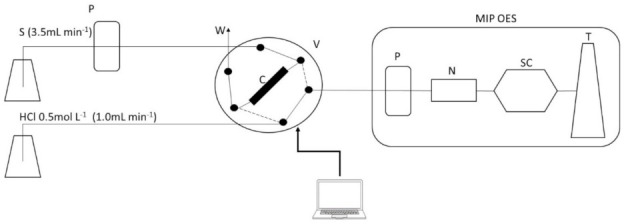
FI system for the on-line preconcentration coupled to MIP OES. P: peristaltic pump; V: 6-port injection valve; C: minicolumn; W: waste; S: sample or standard; HCl: hydrochloric acid for elution; N: nebulizer; SC: spray chamber; and T: torch. Dotted lines represent the valve position during the elution step and solid lines represent the position during the preconcentration step.

**Figure 4 molecules-26-01662-f004:**
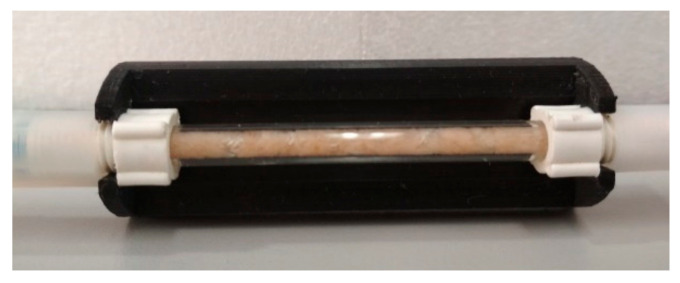
Minicolumn packed with sisal fiber.

**Table 1 molecules-26-01662-t001:** Operating conditions microwave-induced plasma optical emission spectrometer (MIP OES).

	Selected by the Commercial Software ^1^		Selected for This Method	
	Cu	Mn	Cu	Mn
Viewing position	10	−10	10	−10
Nebulizer flow (L min^−1^)	0.65	0.95	0.9	0.9

^1^ When the nebulizer flow changes conditions, due to default optimal values established by the manufacturers, the stabilization time needed causes imprecision.

**Table 2 molecules-26-01662-t002:** Central composite design (3-level, 3 variables).

Exp#	Preconcentration Time (s)	pH	(HCl) (mol L^−1^)	Cu (Peak Height ^1^)	Mn (Peak Height ^1^)
1	30	4	0.2	674	127
2	30	4	0.5	589	114
3	30	6	0.2	401	99
4	30	6	0.5	363	93
5	60	5.5	0.3	1289	226
6	90	4	0.2	1872	277
7	90	4	0.5	1798	258
8	90	6	0.2	1079	223
9	90	6	0.5	1049	198
10	90	5.5	0.3	1854	298
11	90	5.5	0.5	1925	321

^1^ Peak height values refer to intensity of the analytical signal (emission).

**Table 3 molecules-26-01662-t003:** Figures of merit for the flow injection (FI) FI-MP OES method.

	Cu	Mn
Linear range (µg L^−1^)	12–500	30–500
LOD (µg L^−1^)	3.7	9.0
LOQ (µg L^−1^)	12	30
Trueness (%R) *	96.5	98.0
Precision (%RSD, *n* = 6) **	4.3	2.6
Precision (%RSD, *n* = 6) ***	9	10

* mean ± standard deviation, *n* = 6; ** using spiked samples; *** using real water samples. LOD: limit of detection; LOQ: limit of quantification.

**Table 4 molecules-26-01662-t004:** Comparison of Cu and Mn levels in sample waters using ETAAS.

	Developed Method (µg L^−1^)		ETAAS (µg L^−1^)		*t* Value	
Sample	Cu	Mn	Cu	Mn	Cu	Mn
Well water 1	93 ± 9	ND	82 ± 1	<LOQ	2.44	--
Well water 2	79 ± 3	ND	81 ± 1	<LOQ	1.50	--
Well water 3	108 ± 4	175 ± 9	114 ± 2	161 ± 1	−1.07	2.2
Well water 4	124 ± 11	ND	113 ± 1	<LOQ	2.03	--
Well water 5	61 ± 5	ND	55 ± 2	<LOQ	1.14	--
Well water 6	35.4 ± 2.4	57.5 ± 1.3	39.5 ± 5.0	56.9 ± 1.0	−0.44	0.37
Tap water	106 ± 1	ND	105 ± 1	<LOQ	0.68	--
Rainwater	ND	ND	<LOQ	<LOQ	--	--

Mean ± standard deviation. Student’s *t*-test (0.05, 4) = 2.78; LOQ (ETAAS) 4.5 and 1.4 µg L^−1^ for Cu and Mn, respectively. ND: not detected.

**Table 5 molecules-26-01662-t005:** Operation conditions for ETAAS determinations.

Stage	Temperature (°C)	Hold Time (s)	Ramp Rate (°C s^−1^)	Internal Ar Flow (mL min^−1^)
Drying	110	1	30	250
Drying	130	15	20	250
Pyrolysis	1100^(Cu)^/1050^(Mn)^	10	20	250
Atomization	2300^(Cu)^/2100^(Mn)^	0	5	0
Cleaning	2600	1	5	250

## Data Availability

All the data is reported in this manuscript.

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
