# Peer review of "On-Line Preconcentration and Simultaneous Determination of Cu and Mn in Water Samples Using a Minicolumn Packed with Sisal Fiber by MIP OES"

_molecules, 2021, doi:10.3390/molecules26061662_

Round 1

Reviewer 1 Report

The development of the novel method of determination of Cu and Mn in waters on the trace level is presented in a clear way. The authors showed that the method has a good potential and is indeed effective. One of its advantages is exploitation of biosorbent, instead of commercially produced materials, which is more respectful to the environment. Overall, the manuscript is in a good shape (some editing in the text composition might be needed - I see to many paragraphs) and I suggest only a few corrections / improvemenets:

L31-43: I would merge the first three paragraphs into a single one.
L57: u L-1 ?
L123: hight ?
L127: Figure 1: It could be interesting for a reader to add an information about the concentration of the measured samples showed, either in the figure or caption.
L128-130: The sentence is hard to understand, please rephrase.
L135: Table 1: Consider adding more operating conditions (in another table?), it might be helpful for someone working with the MIP OES.
L179-181: Again, the sentence is hard to understand, please rephrase.
L186: Figure 2: The same recommendation as for Figure 1.
L225: advantajes ?
L279: ...by dos Santos et al. - add ref. number here
L298: C° ?
L306: hidroxide ?
L323 ...and no other... - please rephrase

Reviewer 2 Report

The authors of the present manuscript should take in consideration the following points:

-Lines 123-124 and Figure 1. The authors declared that signal height in replicates is relatively imprecise. Nevertheless, it seems that a decrease of the signal height is accompanied by a peak braodening. Why do not considering the peak area rather than its height?

-Paragraph 2.2. Apparently, the authors used an experimental design to identify a number of experimental conditions in which the responses (heights of Cu and Mn signals were simply compared). This seems a reductive application of the Design of Experiments. Actually a statistical model describing  each of the two responses could be built, which allows prediction of the responses in the whole experimental domains and not only in the points of the design.  Why not taking benefit of this possibility?

-Table 3. How the authors explain the fact that precision in the analysis of real samples is lower than that of spiked samples?

-Table 4. Please explain the meaning of ND.

-Lines262-268. Please specify whether ETAAS measurements are performed on raw or pre-concentrated water samples. Which is the sample volume?

Reviewer 3 Report

Very interesting themathic about metal pollution detection with a novel method, ample introduction, good methodology, i appreciate the novelty of your idea, the results are cleraly presented, numerous refercences are used, the ENglish is good. Please further discuss the role of your device in reducing the pollution with metals, please reffer to the fallowing article:

Saramasan C, Identification, Communication And Management Of Risks Relating To Drinking Water Pollution In Bihor County, Environmental Engineering and Management Journal, November/December 2008, Vol.7, No.6, 769-77

Round 2

Reviewer 2 Report

The paper can be accepted